# Autophagic- and Lysosomal-Related Biomarkers for Parkinson’s Disease: Lights and Shadows

**DOI:** 10.3390/cells8111317

**Published:** 2019-10-25

**Authors:** Helena Xicoy, Núria Peñuelas, Miquel Vila, Ariadna Laguna

**Affiliations:** 1Neurodegenerative Diseases Research Group, Vall d’Hebron Research Institute (VHIR)-Center for Networked Biomedical Research on Neurodegenerative Diseases (CIBERNED), 08035 Barcelona, Spain; helena.xicoy@vhir.org (H.X.); nuria.penuelas@vhir.org (N.P.); miquel.vila@vhir.org (M.V.); 2Department of Cell biology, Radboud Institute for Molecular Life Sciences, Radboud University Medical Centre, 6525 GA Nijmegen, The Netherlands; 3Department of Molecular Animal Physiology, Donders Institute for Brain, Cognition and Behaviour, Donders Centre for Neuroscience, Faculty of Science, 6525 GA Nijmegen, The Netherlands; 4Department of Biochemistry and Molecular Biology, Autonomous University of Barcelona, 08193 Barcelona, Spain; 5Catalan Institution for Research and Advanced Studies (ICREA), 08010 Barcelona, Spain

**Keywords:** Parkinson’s disease, biomarker, autophagy, lysosome, glucocerebrosidase, alpha synuclein

## Abstract

Parkinson’s disease (PD) is a neurodegenerative disorder that currently affects 1% of the population over the age of 60 years, for which no disease-modifying treatments exist. This lack of effective treatments is related to the advanced stage of neurodegeneration existing at the time of diagnosis. Thus, the identification of early stage biomarkers is crucial. Biomarker discovery is often guided by the underlying molecular mechanisms leading to the pathology. One of the central pathways deregulated during PD, supported both by genetic and functional studies, is the autophagy-lysosomal pathway. Hence, this review presents different studies on the expression and activity of autophagic and lysosomal proteins, and their functional consequences, performed in peripheral human biospecimens. Although most biomarkers are inconsistent between studies, some of them, namely HSC70 levels in sporadic PD patients, and cathepsin D levels and glucocerebrosidase activity in PD patients carrying *GBA* mutations, seem to be consistent. Hence, evidence exists that the impairment of the autophagy-lysosomal pathway underlying PD pathophysiology can be detected in peripheral biosamples and further tested as potential biomarkers. However, longitudinal, stratified, and standardized analyses are needed to confirm their clinical validity and utility.

## 1. Introduction

Parkinson’s disease (PD) is the second most common neurodegenerative disorder after Alzheimer’s disease, affecting 1% of the population over the age of 60 years [1]. With the increase in life expectancy that our society has experienced in the last decades, the incidence and prevalence of PD is expected to increase substantially, which represents a major economic and social burden [2]. Typical PD symptoms include motor (i.e., resting tremor, movement slowness, rigidity, and postural instability) and non-motor features (i.e., hyposmia, sleep disorders, autonomic dysfunction, neuropsychiatric alterations, and sensory symptoms) [3,4]. Currently, diagnostic criteria for PD require the identification of the classical motor symptoms [5], which are first noticeable when there is already 40–60% neuronal loss in the substantia nigra (SN) [6]. The current treatments for PD, both medications and surgery, are far from disease-modifying and remain symptomatic since they aim at controlling the motor signs caused by dopamine deficiency.

One of the main barriers for the development of treatments able to modify disease progression is the advanced stage of neurodegeneration at the moment of diagnosis. Hence, the identification of early diagnostic PD biomarkers able to accurately diagnose the disease years or decades before the manifestation of major clinical features is fundamental. Biomarkers for PD could be used not only for early diagnosis (diagnostic marker) but also to predict the risk of PD (risk marker), predict disease progression (prognostic markers), describe disease severity (staging markers), support treatment choice (theragnostic markers), and evaluate treatment response (response markers). Research on biomarkers in PD has increased substantially over the past five years, and several systematic reviews and meta-analyses on this topic have been published [7,8,9,10,11].

Identifying a successful biomarker, as well as finding effective neuroprotective therapies, depends inevitably on fully understanding the pathophysiology underlying the disease. At the neuropathological level, PD is mainly characterized by the loss of SN dopaminergic neurons and the presence, in affected regions, of alpha-synuclein (a-SYN)-containing intracytoplasmic inclusions called Lewy bodies [12]. Although the etiology of PD is still unknown, several molecular pathways have been extensively associated with the pathology, including mitochondrial function, oxidative stress, endoplasmic reticulum stress, immune response, proteasome system, and the autophagy-lysosomal pathway (ALP) [13]. In particular, ALP plays a pivotal role in the cellular protein quality control system, and its activity is relevant for maintaining the homeostasis of neurons by degrading and recycling a diversity of complex molecules and long-lived dysfunctional organelles [14] (Figure 1). The ALP is a multistep process that begins with the activation of autophagy, a finely regulated process that contributes to reroute cytoplasmic components toward the lysosome for degradation. This process can be activated by different cellular stress conditions like nutrients deprivation [14], endoplasmic reticulum stress derived from accumulated misfolded proteins [15], or viral infections [16]. The most well-established type of autophagy is known as macroautophagy, in which double-membrane vesicles called autophagosomes are non-specifically loaded with material destined to lysosomal degradation [14]. Another form of autophagy known as chaperone-mediated autophagy (CMA) allows for the selective proteolysis of specific proteins bearing a KFERQ-like sequence, such as a-SYN [17,18]. ALP culminates with the fusion of autophagosomes with lysosomes, which are specialized single-membrane vesicles that contain several hydrolytic enzymes [14]. Because of its essential function, ALP dysfunction is involved in various neurodegenerative diseases such as PD, dementia with Lewy bodies, Alzheimer’s disease, frontotemporal dementia, and Huntington’s disease [14,19]. In addition, the importance of ALP impairment in PD is strengthen by the fact that it represents one of the major routes for the intracellular degradation of a-SYN [20,21,22,23], evidenced by (i) the presence of a-SYN inclusions in the brain tissue of patients with lysosomal storage disorders [24] and (ii) the accumulation of a-SYN insoluble forms in cellular and animal models with impaired ALP, including PD patient–isolated fibroblasts, induced pluripotent stem cells (iPSC)-derived dopaminergic neurons, and mouse models [22].

The importance of ALP impairment in PD is further supported by genetic studies reporting (i) familial genes involved in ALP (i.e., *GBA*, *LRRK2*, *SNCA*, *ATP13A2*, *VPS35*, *FBXO7,*
Table 1) [25], (ii) an enrichment of lysosomal genes associated with the risk of PD development in genome wide association studies [26,27], and (iii) mutations in *GBA*, which encodes for the lysosomal hydrolase glucocerebrosidase (GCase) [28], as the most common genetic risk factor for PD [25,29,30,31]. Homozygous *GBA* mutations cause an inherited autosomal recessive lysosomal storage disorder, known as Gaucher disease (GD), characterized by a loss of function of the GCase protein, leading to accumulation of its substrate, the glucosylceramide, in different organs, such as the liver, the blood, the spleen, the lungs, and the nervous system [32]. In GD patients, neurological and motor impairment have been reported together with a-SYN positive inclusions detected in brain tissue [33]. The relevance of PD associated to GCase dysfunction resides in multiple factors. First, 7–10% of PD patients considered as sporadic (sPD) are actually carriers of *GBA* mutations (GBA-PD) [30,34,35]. Second, up to 1% of individuals in the general population are asymptomatic (non-manifesting) heterozygous *GBA* mutation carriers (GBA-NMC) [36], having a 20-fold increased risk to develop PD as compared to non-carriers [37,38,39,40]. Third, GBA-PD patients usually manifest an earlier onset and a more severe course of the parkinsonism compared to those PD patients not carrying *GBA* mutations, in particular with greater cognitive decline, faster progression and increased risk of mortality [41,42,43,44,45,46,47]. Fourth, altered GCase activity has also been reported in other genetic cohorts (i.e., PD patients carrying *LRRK2*, *PRKN* and *SNCA* mutations) and even in sPD patients [48,49,50]. Overall, *GBA* mutation carriers represent a relatively frequent, well defined and stratified cohort responsive to a-SYN modulating therapies and also to GCase activity therapies [51]. Moreover, biomarkers monitoring the function of the ALP and therapeutic approaches modulating this pathway could potentially be of relevance to a wide range of PD patients and subjects at-risk.

Thus, defining biomarkers to monitor the function of the ALP is of relevance (i) because of the existing pathophysiological link with PD and (ii) because genetic cohorts of *GBA* mutation carriers are currently in the spotlight as one of the most promising prodromal PD cohorts to use in future clinical trials for disease-modifying therapeutic interventions. In this review, we focused on proteins related to the ALP function and on the consequences of their dysfunction, and discussed whether they may serve as molecular and biochemical biomarkers for PD in biological fluids or human-derived cells. We searched PubMed up to 21 June 2019 and retrieved references from relevant articles obtained using a broad custom search criterion (Appendix A).

## 2. Expression of Autophagic-Lysosomal Proteins

The detection of autophagic and lysosomal proteins in peripheral human biospecimens may provide a read-out of lysosomal dysfunction in PD and holds promise for the development of PD biomarkers.

### 2.1. Autophagosome Formation–Related Proteins

The microtubule-associated protein 1A/1B light chain 3B (hereafter referred to as LC3) plays a central role in autophagosome membrane structure and is the most widely used marker of autophagosomes [68]. Data on LC3 levels as PD biomarkers is diverse and not always consistent between different studies. LC3 levels in the cerebrospinal fluid (CSF) of PD patients were described both lower [69] and not changed [70] compared to controls. In the same line, one study showed no differences in the LC3II/LC3I ratio [71], while another study showed increased LC3 protein levels in peripheral blood mononuclear cells (PBMCs), although with no correlation with clinical variables [72]. Also, higher levels of LC3 mRNA and protein levels were found in leukocytes from sPD patients [73]. Different results were again reported when using dermal fibroblasts derived from sPD patients. One study showed decreased LC3 levels [74], while other studies reported no differences [75,76]. There are no consistent results in studies focusing on genetic PD cohorts either. GBA-PD fibroblasts showed no alterations of LC3 levels [77], while iPSC-derived dopaminergic neurons from GBA-PD patients carriers of the N370S mutation (N370S-GBA PD) showed higher LC3 protein levels [50]. Fibroblasts from PD patients with *PRKN* mutations (PRKN-PD) showed increased levels of LC3 in a study assessing only one patient [78], while decreased levels of LC3 were reported in another study assessing seven PRKN-PD patients [79]. Finally, in PD patients with *LRRK2* mutations (LRRK2-PD), fibroblasts carrying different mutations (i.e., R1441G, Y1699C, G2019S) showed no differences in LC3 levels [80], although one study in fibroblasts from G2019S-LRRK2 PD patients showed increased LC3I to LC3II conversion in the majority of patients, suggesting an increased formation of autophagosomes [81].

Similarly, data related to other autophagy regulators besides LC3 is not more revealing. Lower CSF levels of Beclin-1 and ATG5 were observed in sPD patients [69]. Transcripts known to participate in the autophagosome formation pathway (i.e., *ULK3*, *ATG2A*, *ATG4B*, *ATG5*, *ATG16L1*, *HDAC6*) were found downregulated in PBMCs from sPD patients [71]. In the same study, protein levels of ULK1, Beclin-1, and autophagy/beclin-1 regulator 1, which are all involved in phagophore formation, were significantly increased in PBMCs from sPD patients [71]. Interestingly, although the protein levels of these proteins did not correlate with the clinical severity of PD, they did correlate with the levels of oligomeric a-SYN in these cells [71]. Finally, Beclin-1 protein levels were also found increased in iPSC-derived dopaminergic neurons from N370S-GBA PD patients [50].

### 2.2. Autophagic Cargo Delivery–Related Proteins

Sequestosome-1, also known as the ubiquitin-binding protein p62, is an autophagosome cargo protein that loads ubiquitin-tagged misfolded proteins for selective autophagy. Protein levels of p62 have also been extensively analyzed as putative biomarkers for PD, but again, results seem to be inconsistent for both sporadic and genetic PD cases. p62 levels in sPD-derived fibroblasts were decreased in one study [74] and not changed in another [76]. No alterations of p62 were found in GBA-PD–derived fibroblasts [77], while PRKN-PD–derived fibroblasts consistently showed lower levels of p62 protein [78,79]. In LRRK2-PD–derived fibroblasts, one study assessing different *LRRK2* mutations (i.e., R1441G, Y1699C, G2019S) showed decreased levels of p62 specifically in R1441G and Y1699C mutant LRRK2-PD–derived fibroblasts [80], while another study including only G2019S-LRRK2 PD patients found increased p62 levels [81]. The discrepancy between both studies may be explained by the different phenotypic display of specific *LRRK2* mutants [82]. Finally, iPSC-derived dopaminergic neurons from N370S-GBA PD patients showed increased p62 protein levels compared to controls [50].

Protein levels of HSC70, a member of the heat shock protein 70 family of chaperones that transfers the substrate protein to the lysosomal membrane in the process of CMA, have been consistently found decreased both at the protein and transcript level in PMBCs from sPD patients [83,84]. Likewise, PMBCs from genetic PD forms, i.e., PD carriers of the A53T *SNCA* mutation (A53T-SNCA PD) and GBA-PD, also showed decreased HSC70 protein levels [83].

### 2.3. Lysosomal Proteins

#### 2.3.1. Structural Membrane Proteins

The Lysosomal-associated membrane proteins 1 and 2 (LAMP1 and LAMP2) are glycoproteins found in the membrane of lysosomes and are partially responsible for maintaining lysosomal integrity, pH, and catabolism [85]. Because of their critical functions and wide expression, they represent the most widely used markers of lysosomes. Although one study found no differences in the CSF levels of LAMP2 in sPD [86], lower CSF levels of LAMP1 [70] and LAMP2 [69,70] were consistently described in sPD and female LRRK2-PD patients [86]. Accordingly, LAMP2a, involved in selective CMA, mRNA, and protein levels, was found to be decreased in leukocytes from sPD patients [73]. In contrast, no differences were found in LAMP2 mRNA and protein levels in PMBCs from sPD patients [84]. Protein levels of LAMP1, LAMP2a, and LIMP2, a GCase-specific lysosomal receptor, were increased in iPSC-derived dopaminergic neurons from N370S-GBA PD patients [50]. Finally, increased blood expression of ATP13A2, a lysosomal cation-transporting ATPase genetically associated with PD [59], has been detected both in treated and untreated PD patients [87].

#### 2.3.2. Enzymes

There are conflicting results regarding GCase levels in different biosamples. At a transcriptional level, GCase mRNA levels were decreased in leukocytes from GBA-PD patients [88] and in skin fibroblasts derived from patients carrying specific *GBA* mutations (L444P and the recombinant gene recNciI), while no differences were found in cases carrying N370S or homozygous E326K *GBA* mutations [77]. At a protein level, both lower GCase levels and higher endoglycosidase-H sensitive GCase fraction [77] or no differences in enzyme levels [75] were reported in skin fibroblasts of GBA-PD patients. Similarly, no differences of total GCase protein levels were observed in iPSC-derived dopaminergic neurons from N370S-GBA PD patients, but an additional GCase isoform of higher molecular weight and a higher endoglycosidase-H sensitive GCase fraction were observed in cells derived from N370S-GBA PD patients [50]. Finally, GCase protein levels were decreased in PMBCs from A53T-SNCA and GBA-PD cases but did not differ in sPD patients [83,89].

Similarly, data is also available for other lysosomal enzymes. Different transcripts of the lysosomal pathway (i.e., *GLA*, *CTSA*, *CTSB*, *CTSD*, *PSAP*, *ASAH1*, *HEXB*) were found enriched in PBMCs from sPD patients [71]. Accordingly, higher protein levels of beta-hexosaminidase, essential for the degradation of GM2 gangliosides, and cathepsin D, a lysosomal aspartyl protease, were found in skin-derived fibroblasts from GBA-PD and GD patients [77]. Higher levels of cathepsin D immature forms were also reported in PRKN-PD-derived fibroblasts [78]. In line with these results, iPSC-derived dopaminergic neurons from N370S-GBA PD patients also showed increased cathepsin D levels [50]. On the contrary, the levels of alpha-galactosidase, a glycoside hydrolase that, when mutated, results in the rare lysosomal storage disorder Fabry disease [90], were found decreased in leukocytes from sPD patients [73].

## 3. Direct Markers of Autophagic-Lysosomal Function

One step beyond the assessment of protein levels is to look at the function of those proteins in peripheral human biospecimens, which may provide a direct functional read-out of the ALP activity to be used in the development of PD biomarkers.

### 3.1. Enzymatic Activities

#### 3.1.1. Glucocerebrosidase

Decreased GCase activity has been consistently detected in dried blood spots from GBA-PD patients [48,91], and it was significantly lower than in non-GBA PD patients [48,91,92]. However, the possible use of GCase activity as a biomarker in sPD peripheral biosamples is more controversial. Some studies reported decreased activity in CSF [49,93,94], serum [94], and dried blood spots [48], while others reported no differences in CSF [95], dried blood spots [91], and lymphocytes [96]. Surprisingly, increased GCase activity was found in dried blood spots of LRRK2-PD patients [48].

In PMBCs, GCase activity was found decreased only in GBA-PD compared to healthy controls, while no differences were found in A53T-SNCA PD or sPD [83]. However, when assessing monocytes specifically, which have been reported to have the highest GCase activity within all peripheral immune cells [97], sPD patients showed decreased GCase activity, although with limited classification ability and showing no correlation with disease severity [89]. Again in monocytes, consistent with previous results in PBMCs, Gcase activity was decreased in GBA-PD patients and in healthy GBA-NMC [89]. Similar results were obtained in leukocytes, another population of PMBCs. Lower GCase activity was reported for both manifest GBA-PD patients and their GBA-NMC healthy relatives [88], while sPD and PRKN-PD patients showed no differences in GCase activity [89,98]. Remarkably, a positive correlation between GCase activity levels in leukocytes and disease duration has been reported in sPD patients [98].

GCase activity has also been widely assessed in skin-derived fibroblasts from different types of PD patients. No differences at baseline were found in sPD patients [75,77], while lower GCase activity was found in GBA-PD fibroblasts and in cases with PD and GD [75,77]. Interestingly, this decreased GCase activity was consistent in all subjects with *GBA* mutations, including GD patients alone and healthy GBA-NMC and manifesting PD mutation carriers [77]. Also, in iPSC-derived dopaminergic neurons from N370S-GBA PD patients, GCase activity was decreased compared to controls [50].

#### 3.1.2. Galactosidases

The activity of alpha-galactosidase in dried blood spots was found to be either reduced [99] or not different [91,100] in sPD patients, and decreased in GBA-PD [91]. As occurred for GCase activity, elevated alpha-galactosidase activity was reported in LRRK2-PD [99]. Interestingly, alpha-galactosidase is encoded by an X-linked gene, and it has been described to be only altered in females when both genders are analyzed separately [99]. The activity of beta-galactosidase is again controversial, being reported in some sPD studies as increased in serum [100] and CSF [95], while others reported no changes in serum [94] and CSF [93,94]. Only one study assessed the ratio beta-galactosidase/alpha-galactosidase activity and found it increased in serum samples from PD patients [100]. No differences in alpha-galactosidase and beta-galactosidase activity were found either in sporadic or genetic PD fibroblasts [75,77]. Conversely, lower alpha-galactosidase activity was reported when leukocytes from sPD patients were analyzed [101].

#### 3.1.3. Other Lysosomal Enzymes

Although one study reported increased beta-hexosaminidase activity in the CSF of sPD patients [93], most studies found no differences in activity neither in serum [94,100], nor in CSF [49,94,95]. Similarly, one study reported decreased CSF activity of alpha- and beta-mannosidase in sPD patients [94], but other studies found no differences either in serum [94] or in CSF [93,95]. Accordingly, no differences in hexosaminidase or manosidase activities were reported in sPD leuckocytes [98,101], as well as in sPD and GBA-PD skin-derived fibroblasts [75]. Furthermore, cathepsin D activity in the CSF has been described as decreased and as unaltered in different studies [49,95], while increased cathepsin E activity was found in the CSF of sPD patients [95]. Also, acid sphingomyelinase activity was increased in dried blood spots of LRRK2-PD patients [99] but not different in sPD [91]. Finally, the activity of some enzymes has been studied only once, namely the lack of differences in galactocerebrosidase, alpha-glucosidase, or alpha-iduronidase activity in dried blood spots of sPD patients [91] and the decreased alpha-fucosidase activity in CSF of sPD patients [95].

### 3.2. Non-Enzymatic Assays

Different assays consistently reported autophagic alterations in PD-derived cells. In PBMCs from sPD patients, ^3^H-leucine pulse-chase experiments showed significantly reduced lysosomal activity including macroautophagy and CMA [102]. Higher autophagic flux assessed by pH-sensitive fluorescent viral constructs was shown in sPD and G2019S-LRRK2 PD–derived skin fibroblasts and confirmed by electron microscopy with an increase in the number of autophagic vesicles observed [74]. PRKN-PD–derived fibroblasts also showed increased lysosomal compartments stained with lysosomal fluorescent markers but decreased lysosomal function markers such as RAB7A and ATP6V1G1, and decreased proteolytic activity [78]. iPSC-derived dopaminergic neurons from N370S-GBA PD patients presented higher number of lysosomes per cell and higher lysosomal area determined by electron microscopy studies [50]. On the contrary, decreased lysosomal compartment (decreased number of lysosomes, lysosomal area, and mean size) has been shown in neural stem cells derived from G2019S-LRRK2 PD patients [103], though these measurements were performed analyzing lysosomal mass by immunostaining of LAMP2.

## 4. Indirect Markers of Autophagic-Lysosomal Function

Because of the diverse and crucial range of functions attributed to lysosomes, failure of lysosomes leads to the accumulation of dysfunctional organelles and proteins, such as a-SYN [23], as well as to alterations in lipid homeostasis [104].

### 4.1. A-SYN Levels

A-SYN is an endogenous presynaptic protein believed to function in neurotransmitter release [105]. Abnormal accumulation and aggregation of a-SYN, in part due to defects in protein quality control mechanisms such as autophagic clearance, has long been thought to be a pathogenic cause of PD [23]. Thus, the levels of total a-SYN and of different a-SYN conformations have long been under focus for PD biomarker development. We do not aim to review all these studies (for extensive information on this subject, see [11,21]), but instead, only focus in the studies using GBA-PD cohorts where GCase activity seemed consistently decreased and a-SYN levels increased.

Higher oligomeric a-SYN levels were found in plasma of GBA-PD patients [91], while no changes were identified in plasma of healthy GBA-NMC [106]. A recent study reported increased serum levels of a-SYN in those GBA-NMC with higher combined clinical risk scores [46], suggesting that monitoring of a-SYN levels could be predictive of phenoconversion in GBA-NMC. Another study reported higher a-SYN dimer species in erythrocyte membranes from GBA-PD patients, although these measurements did not correlate with disease severity [107]. Interestingly, GD erythrocyte membranes also presented higher levels of a-SYN dimerization [108]. On the contrary, one study found no correlation between lysosomal hydrolases in dry blood spots and a-SYN levels in plasma of GBA-PD patients [91]. Finally, higher levels of extracellular a-SYN were found in iPSC-derived dopaminergic neuronal cultures from N370S-GBA PD patients, while intracellular a-SYN levels were unaltered [50]. A defective ALP has been observed in iPSC-derived dopaminergic neurons from GD and PD individuals carrying *GBA* mutations, which may account for the increased levels of a-SYN found in these neurons [50,109].

### 4.2. Mitophagy Markers

PD has long been associated with mitochondrial dysfunction [110]. Impaired autophagy results in a disruption of mitophagy, which is the process of targeted degradation of non-functional mitochondria [111]. Based on the pathophysiology of the disease, the identification of abnormal mitophagy seemed a reasonable source of possible biomarkers in peripheral biospecimens, although the existing evidence is ambiguous.

Increased mitophagy markers (PRKN levels, cleaved PINK1 levels, and co-localization of TOM20 and LAMP1) were reported in sPD-derived fibroblasts [76] and when subdividing sPD fibroblasts according to their vulnerability to the mitochondrial depolarizing agent valinomycin [112]. This last study also showed increased mitophagy markers in LRRK2-PD, higher levels of PRKN and cleaved PINK only in G2019S-LRRK2 PD and co-localization of TOM20 and LAMP1 in both G2019S- and R1441C-LRRK2 PD [112]. In contrast, other studies reported decreased mitophagy in G2019S-LRRK2 PD and PRKN-PD–derived fibroblasts [113], in G2019S-LRRK2 PD–derived neural stem cells [103], and in starved PD–derived dermal fibroblasts from *DJ1* mutation carriers [114].

### 4.3. Lipid Levels Influenced by Autophagy-Related Proteins

Altered lipid metabolism has been recently associated to PD [104]. GCase is a lysosomal enzyme involved in sphingolipid metabolism, as it hydrolyzes the glycolipids glucosylceramide (GlcCer) and glucosylsphingosine. GCase catabolizes GlcCer to glucose and ceramide, which is recycled to generate new glycosphingolipids and sphingomyelins [115]. When GCase is dysfunctional, GlcCer and other lipids accumulate in the lysosomes of many tissues, thereby compromising lysosomal function, and possibly contributing to PD pathogenesis by affecting microglial activation and neuronal damage [116].

Increased levels of plasma ceramides, monohexylceramides and lactosylceramides were found in sPD patients compared to controls [117], while another study showed decreased plasma levels of 36 out of 59 ceramide species assessed, and a positive correlation with GCase activity in monocytes from the same patients [89]. GBA-PD patients had increased levels of hexosylsphingosines and lysosphingomyelin in dried blood spots when compared to controls [118], and increased serum monohexosylceramide, ceramide, and sphingomyelin, together with decreased phosphatidic acid, phosphatidylethanolamine, plasmalogen phosphatidylethanolamine, and acyl phosphatidylglycerol when compared to non-GBA PD patients [119]. In iPSC-derived dopaminergic neurons from N370S-GBA PD, levels of ceramide 20:0 were reduced and levels of ceramide 16:0 and 24:0 were increased, while no differences were found in total GlcCer accumulation, compared to controls [50]. Finally, PRKN-PD-derived skin fibroblasts showed increased levels of phosphatidylinositol, phosphatidylserine, lysophosphatidylcholine, and specific glycosphingolipid species (GM2 and GM3 gangliosides) [120].

## 5. Discussion

The aim of this review was not to exclusively highlight those biomarkers with the best predictive capacity for PD but to present for the first time in a systematic way the different studies that have been performed searching for biomarkers in relation to a biological pathway critical for PD pathophysiology, namely the ALP. Although usual biomarker reviews focus primarily in CSF and blood-derived biomarkers for their possible clinical utility as diagnostic biomarkers, we were interested in having a broader picture of the scenario. In this way, these could be used not only in clinical diagnosis but also, for instance, to monitor target engagement in drug development and outcome measures in clinical trials. Thus, we considered biomarkers as any biochemical or molecular event detectable in human peripheral biosamples and able to provide information on disease condition.

The results from this study are inconclusive because most of the data is ambiguous and/or not consistent between studies (Table 2). Severe inconsistency between the results found in different studies for autophagic-lysosomal markers (e.g., LC3, p62, LAMP1/2) exclude them so far from being possible reliable biomarkers. The exceptions might be the levels of HSC70 and cathepsin D, since the available results seem consistent, although the number of studies is limited. More consistent findings are revealed when searching exclusively for biomarkers in genetic PD cohorts (Table 3). These observations highlight the fact that, despite the fact that most genetic PD cases seem clinically indistinguishable from sPD, the underlying pathogenic mechanisms might be different. Thus, their management must be carefully considered, especially when thinking of disease-modifying therapeutic interventions. An interesting phenomenon worth mentioning is observed in LRRK2-PD cases, where the activities of three lysosomal enzymes (i.e., GCase, alpha-galactosidase and acid sphingomyelinase) were increased, as opposed to decreased or not changed, in GBA-PD and sPD patients, respectively [99]. All in all, genotyping for at least the most common PD-associated mutations and variants should be considered for decisions on clinical trial enrollment and possibly on the daily clinical practice as well.

Nevertheless, there might be some light between the shadows. One of the most reliable and encouraging biomarkers is decreased GCase activity, which was consistent in all subjects with *GBA* mutations, including GD patients, GBA-NMC individuals, and GBA-PD patients (Table 3). GCase activity was also found to be significantly reduced in different studies comparing PD patients, independent from the presence of *GBA* mutations, with healthy or neurological controls [48,121,122]. These data are in agreement with the decreased GCase activity repeatedly found in postmortem brain tissues [121,122,123,124,125]. However, one study reported no significant differences between groups in GCase activity [95]. The variability in findings from biomarker studies might reflect (i) the heterogeneous characteristics of the selected cohorts and (ii) different procedures for biosamples collection and posterior analysis of biomarkers, since lysosomal enzyme activities are highly influenced by pre-analytical factors [11]. Therefore, these findings highlight the need for more in-depth investigations with the following characteristics to confirm the diagnostic utility of lysosomal enzymes as biomarkers for PD. First of all, recruitment of larger and longitudinal cohorts. Second, coordination of multicentric studies, to consider possible sociodemographic and other confounding factors [126]. Third, inclusion of patients with prodromal manifestations and/or at risk of developing PD, to determine both their diagnostic accuracy and predictability on clinical outcome. Fourth, inclusion of patients affected by other neurodegenerative disorders, to validate their discriminatory capacity. Finally, standardization of the procedures, as established in [127] or by the Parkinson’s Progression Markers Initiative (PPMI) from the Michael J.Fox Foundation (MJFF), together with their stringent use.

A single biomarker seems unlikely to satisfy all the functions required for a reliable biomarker. Hence, research should aim at defining combinations of various genetic, clinical, biochemical, and imaging biomarkers to strengthen the diagnosis of PD and to allow for future precision medicine in the management of PD. Even within biochemical biomarkers, the data available so far points towards combined biomarkers to increase diagnostic accuracy and predictive capacity. One possibility to explore is the combination of different biomarkers reflecting the pathogenic mechanisms that take place during the course of the diseases. Examples of such are (i) different lysosomal markers, e.g., GCase, cathepsin D and beta-hexaminidase activities [49], beta-galactosidase and alpha-fucosidase activities [95]; (ii) lysosomal markers with a-SYN species, e.g., GCase activity with oligomeric/total a-SYN ratio and older age [93]; and (iii) CSF total a-SYN, amyloid beta 42 and a panel of lysosomal enzymes including GCase [49]. Additionally, current studies on ALP-associated biomarkers for PD lack two non-invasive biosamples used for the diagnosis and monitoring of neurological diseases, namely saliva [128] and urine [129]. It would be interesting to study if those biospecimens can be used to detect some of the biomarkers discussed here. Moreover, some ALP-related parameters, such as lipid droplets [130], have not been explored yet.

Intense research efforts should aim mainly at finding biomarkers to recognize PD before motor features become evident (prodromal) or even to detect an asymptomatic population at risk of developing PD (preclinical). Another important aspect of biomarkers is to differentiate PD from other synucleinopathies (i.e., dementia with Lewy bodies and multiple system atrophy) and other parkinsonian syndromes (i.e., progressive supranuclear palsy and corticobasal degeneration), as misdiagnosis often takes place early in the disease due to substantial clinical overlap and the fact that definitive diagnostic confirmation needs autopsy reports [131].

Currently, clinical trials of a number of neuroprotective compounds targeting the GCase pathway are ongoing, e.g., Ambroxol (NTC02941822) [132]. For the design of these clinical trials, CSF and blood biomarkers as outcome measures are of vital importance, and GCase activity seems so far to be the best candidate [133]. Another priority for these clinical trials aiming at modifying the disease outcome, is the definition of a subgroup of *GBA* mutation carriers at increased risk of converting to manifest PD, which might be achieved by a combination of clinical and molecular biomarkers. Moreover, because of the importance of the ALP pathway for PD independent of *GBA* mutations, one could hypothesize that finding the biomarkers that allow us to predict the conversion of *GBA* carriers to PD may also allow us to reveal those who will develop sPD.

In conclusion, evidence exists that the impairment of ALP underlying PD pathology results in altered protein levels and enzymatic activities that can be detected and quantified in peripheral biosamples. These changes could be further tested as potential biomarkers for PD, probably in combination with other biomarkers. However, longitudinal and standardized analyses are needed to confirm their clinical validity and utility.

## Figures and Tables

**Figure 1 cells-08-01317-f001:**
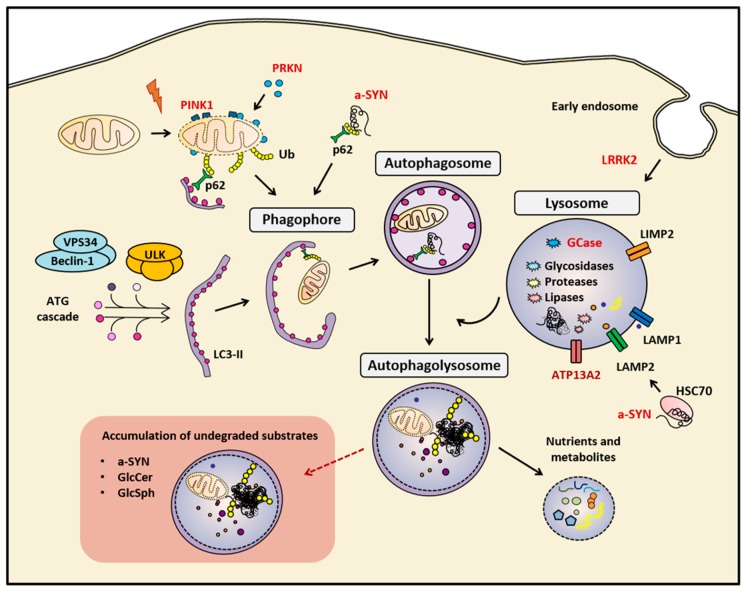
Summary of the main mentioned markers of the autophagy-lysosomal pathway (ALP). The correct function of the ALP is essential for maintaining cellular homeostasis. In cases of genetic or sporadic Parkinson’s disease (PD), the impairment of this biological process leads to the accumulation of undegraded cellular components, such as protein aggregates, and dysfunctional organelles, such as mitocondria. Red-colored proteins are genetically linked to PD. a-SYN (alpha-synuclein), Autophagy proteins (ATG), Cation-transporting ATPase 13A2 (ATP13A2), E3 ubiquitin-protein ligase parkin (PRKN), Ubiquitin (Ub), Glucosylceramide (GlcCer) and Glucosylsphingosine (GlcSph), Heat-shock cognate protein of 70 kilodalton (HSC70), LC3-phosphatidylethanolamine conjugate (LC3-II), Leucine-rich repeat kinase 2 (LRRK2), Lysosomal hydrolase glucocerebrosidase (GCase), Lysosomal integral membrane protein-2 (LIMP-2), Lysosomal-associated membrane protein 1 (LAMP1), Lysosome-associated membrane protein 2 (LAMP2), Phosphatase and tensin homolog (PTEN)-induced kinase 1 (PINK1), Ubiquitin-binding protein p62 or Sequestosome 1 (p62), Unc-51-like kinase (ULK), Vacuolar sorting protein 34 (VPS34)

**Table 1 cells-08-01317-t001:** Summary of familial PD mutations and their ALP involvement. This table summarizes the genes for which mutations have been associated with familial PD [31] either with an autosomal dominant (AD), autosomal recessive (AR), or X-linked dominant (XL-D) mode of inheritance. The table includes the gene symbol, name and aliases, the mode of inheritance and the current knowledge on their involvement in the autophagy-lysosomal pathway (ALP).

Symbol	Name	Aliases	Inheritance	ALP involvement
*LRRK2*	Leucine rich repeat kinase 2	*PARK8; RIPK7; ROCO2; AURA17; DARDARIN*	AD/AR	Yes [52]
*SNCA*	Synuclein alpha	*PD1; NACP; PARK1; PARK4*	AD	Yes [53]
*VPS35*	VPS35 retromer complex component	*MEM3; PARK17*	AD	Yes [54]
*ATXN2*	Ataxin 2	*ATX2, SCA2, TNRC13*	AD	Unknown
*GCH1*	GTP cyclohydrolase 1	*DYT14, DYT5, DYT5a, GCH, GTP-CH-1, GTPCH1, HPABH4B*	AD	Unknown
*PRKN*	Parkin RBR E3 ubiquitin protein ligase	*AR-JP, LPRS2, PARK2, PDJ*	AR	Yes [55]
*PINK1*	PTEN induced kinase 1	*BRPK, PARK6*	AR	Yes [55,56]
*PARK7*	Parkinsonism associated deglycase	*DJ1; DJ-1; GATD2; HEL-S-67p*	AR	Yes [57,58]
*ATP13A2*	ATPase cation transporting 13A2	*CLN12; KRPPD; PARK9; SPG78; HSA9947*	AR	Yes [59]
*PLA2G6*	Phospholipase A2 group VI	*GVI; PLA2; INAD1; NBIA2; iPLA2; NBIA2A; NBIA2B; PARK14; PNPLA9; CaI-PLA2; IPLA2-VIA; iPLA2beta*	AR	Yes [60]
*FBXO7*	F-box protein 7	*FBX; FBX7; PKPS; FBX07; PARK15*	AR	Yes [61]
*DNAJC6*	DnaJ heat shock protein family (Hsp40) member C6	*DJC6; PARK19*	AR	Yes [62]
*SPG11*	SPG11 vesicle trafficking associated, spatacsin	*ALS5; CMT2X; KIAA1840*	AR	Yes [63]
*SYNJ1*	Synaptojanin 1	*EIEE53; INPP5G; PARK20*	AR	Yes [64,65]
*VPS13C*	Vacuolar protein sorting 13 homolog C	*PARK23*	AR	Yes [66]
*RAB39B*	RAB39B, member RAS oncogene family	*WSN; BGMR; WSMN; MRX72*	XL-D	Yes [67]

**Table 2 cells-08-01317-t002:** Summary of the sPD biomarkers. This table includes only those biomarkers assessed in at least two studies. Directionality in sPD compared to controls is depicted (↑, increased; ↓, decreased; ND, no differences). CSF, cerebrospinal fluid; PBMCs, peripheral blood mononuclear cells.

Biomarker	Measurement	Biospecimen	Directionality
alpha/beta-mannosidase	Activity	CSF	↓ [94]/ND [93,95]
		Fibroblasts	ND [75]
		Leukocytes	ND [73]
		Serum	ND [94]
alpha-galactosidase	Activity	Dried blood spots	↓ [99]^/^ND [91,100]
		Fibroblasts	ND [75,77]
		Leukocytes	↓ [73]
	Protein	Leukocytes	↓ [73]
Beclin 1	Protein	CSF	↓ [69]
		PBMCs	↑ [71]
beta-galactosidase	Activity	CSF	↑ [95]^/^ND [93,94]
		Fibroblasts	ND [75,77]
		Serum	↑ [100]/ND [94]
beta-hexosaminidase	Activity	CSF	↑ [93]/ND [49,94,95]
		Fibroblasts	ND [75]
		Leukocytes	ND [98,101]
		Serum	ND [94,100]
Cathepsin D	Activity	CSF	↓ [49]/ND [95]
Ceramides	Lipid	Plasma	↑ [117]/ND [89]
GCase	Activity	CSF	↓ [49,93,94]/ND [95]
		Dried blood spots	↓ [48]^/^ND [91]
		Fibroblasts	ND [75,77]
		Lymphocytes	ND [96]
		Monocytes	↓ [89]
		PBMCs	ND [107]
		Serum	↓ [94]
HSC70	mRNA	PBMCs	↓ [83,84]
LAMP2	mRNA	PBMCs	ND [84]
	Protein	CSF	↓ [70]/ND [86]
		PBMCs	ND [84]
LC3	mRNA	Leukocytes	↑ [73]
	Protein	CSF	↓ [69]/ND [70]
		Fibroblasts	↓ [74]/ND [75,76]
		Leukocytes	↑ [73]
		PBMCs	↑ [72]/ND [71]
p62	Protein	Fibroblasts	↓ [74]/ND [76]
PRKN	Protein	Fibroblasts	↑ [76,112]

**Table 3 cells-08-01317-t003:** Summary of the genetic PD biomarkers. This table includes only those biomarkers assessed in at least two studies. Directionality in genetic PD compared to controls is depicted (↑, increased; ↓, decreased; ND, no differences). CSF, cerebrospinal fluid; EC, extracellular; PBMCs, peripheral blood mononuclear cells; iPSC, induced pluripotent stem cells; DA, dopaminergic.

GBA-PD			
Biomarker	Measurement	Biospecimen	Directionality
a-SYN	Protein—Dimers	Erythrocyte membranes	↑ [107]
	Protein—EC	iPSC-derived DA neurons	↑ [50]
	Protein—Oligomeric	Dried blood spots	↑ [91]
Cathepsin D	Protein	Fibroblasts	↑ [77]
		iPSC-derived DA neurons	↑ [50]
GCase	Activity	Dried blood spots	↓ [48,91]
		Fibroblasts	↓ [75,77]
		iPSC-derived DA neurons	↓ [50]
		Monocytes	↓ [88]
		PBMCs	↓ [83]
	mRNA	Fibroblasts	↓ [77]
		Leukocytes	↓ [88]
	Protein	Fibroblasts	↓ [77]/ND [75]
		iPSC-derived DA neurons	ND [50]
LC3	Protein	Fibroblasts	ND [77]
		iPSC-derived DA neurons	↑ [50]
p62	Protein	Fibroblasts	ND [77]
		iPSC-derived DA neurons	↑ [50]
LRRK2-PD			
LC3	Protein	Fibroblasts	↑ [78,81]/ND [80]
p62	Protein	Fibroblasts	↑ [81]/ND [80]

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
