# Peer review of "Autophagic- and Lysosomal-Related Biomarkers for Parkinson’s Disease: Lights and Shadows"

_cells, 2019, doi:10.3390/cells8111317_

Round 1
Reviewer 1 Report
In general the manuscript reads well. The language is excellent. The authors succeeded in reviewing the sources of autophagy and lysosomal related biomarkers in PD. This manuscript is very well written and scientifically informative. My only suggestion would be to include a sentence or two that will summarize each chapter with authors’ critical view on each presented topic.
Author Response
Reviewer 1
1) My only suggestion would be to include a sentence or two that will summarize each chapter with authors’ critical view on each presented topic.
Answer: The authors completely agree with the reviewer’s suggestion and were actually willing to conclude each chapter with a conclusive sentence that summarized the evidence found for a specific biomarker even before submission. However, because the results for each chapter, considering the different biomarkers in different biosamples, were found inconclusive, the authors decided it was hard to include a summarizing sentence for each chapter without this been repetitive and non-informative. Hence, in order to avoid repetition, the authors favored the option of having two tables summarizing the major findings (Table 2,3), and of writing a final conclusion highlighting the most promising findings and discussing the discrepancy between studies at the end as a general discussion (second paragraph of the discussion section).
Reviewer 2 Report
The authors present a very interesting study on a hot topic of biomarkers for Parkinson’s disease. The topic is insightful and may attract great attention in future research. Moreover, the paper is well written and organized. The manuscript is well structured and sounds scientifically.
The authors comprehensively describe one of the central pathways deregulated during Parkinson’s disease (PD), supported both by genetic and functional studies, namely the autophagy-lysosomal pathway. Graphical visualization is of very good quality. Overall, the manuscript is of a very high quality.
Maybe, I would suggest adding a table of commonly found mutations associated with PD, which authors describe in the introduction. I understand that it is not a direct topic of the study. However, it would add readability for those who are not in the topic.
Author Response
Review 2
1) Maybe, I would suggest adding a table of commonly found mutations associated with PD, which authors describe in the introduction. I understand that it is not a direct topic of the study. However, it would add readability for those who are not in the topic.
Answer: The authors have added a table (Table 1) with the commonly found mutations associated with PD, together with their involvement in the autophagy-lysosomal pathway.
Reviewer 3 Report
In the manuscript "Autophagic- and lysosomal-related biomarkers for Parkinson’s disease: lights and shadows" H Xicoy, N Peñuelas, M Vila and A Laguna reviewed and discussed the ALP-related protein function and consequences of their dysfunction, as well as whether they may be considered as molecular and biochemical biomarkers for PD in biological fluids or human-derived cells. To do that they performed a deep and specific search on PubMed up to June 21st 2019 and retrieved references from relevant articles obtained using a broad custom search criterion (as they reported in Supplemental File).
First of all, as the aim of the work stated in the introduction section “..molecular and biochemical biomarkers for PD in biological fluids..” I wonder why the authors did not mention and discuss the studies on saliva which are recently raising lots of interest.
Line 68: “… or viral infections [14].” The cited article does not discuss any viral infections then authors should amend the sentence, maybe saying something related to the prion-proteins that behave as infection agents but aren’t (that is mentioned in the cited paper).
Line 108: “… GBA-PD patients usually manifest an earlier onset and a more severe course of the disease…” which disease the authors are talking about? They stated in line 104 “The relevance of PD associated with GBA mutations (GBA-PD) resides in multiple factors” but not mention the evidence of association nor cited any reference therefore it is not clear what are they referring to.
Why authors discussed the data on sSIN related only to GBA-PD and did not at least mention the other groups? And only mention the total aSIN despite the oligomeric and phosphorylated forms?
Line 418-420: “…because of the importance of the ALP pathway for PD independently of GBA mutations, one could hypothesize that finding the biomarkers that allow to predict the conversion of GBA carriers to PD may also allow us to reveal those who will develop sPD…” again I found this statement more a speculation than a conclusion with scientific basis.
Moreover, talking about the alterations in lipid metabolism and endo-lysosomal pathways that is actually the topic of this review it should be interesting – and I wonder why authors did not mention that – to discuss the lipid droplets involvement if there is.
At a certain point in the narrative start appearing several acronyms which, in particular for a reader that it is not properly involved in this scientific area, are pretty hard to follow and understand especially cause the lack of any information useful to. For example: LRRK2, PARKN, PRKN and several mutations that are mentioned but without any explanation where they are and provoke therefore I found confusing and disturbing in the reading, it may be better to avoid them unless simple useful info are reported.
Agreeing that the GBA-PD can be considered and represents a good model which can to tell us something more related to the ALP involvement in the PD, but the title and the abstract in this context sound a little bit tricky and pretentious especially when it is stated that “…evidence exists that the impairment of the autophagy-lysosomal pathway underlying pathophysiology..” since the pathophysiology is due to the GBA mutations then the conclusion seems to be sort of biased.
By the way, although I agree with the authors when they clearly stated that it is very importance to conduct further extensive studies focused on finding combinations of biomarkers that allow to predict sPD development, I found this review provides a solid starting point to know what has been done and what can be done to go ahead.
in conclusion, I would be happy to recommend this article suitable to be published if presented in a revised version though I found it pretty well-written and nice to read.
Author Response
Reviewer 3
1) First of all, as the aim of the work stated in the introduction section “…molecular and biochemical biomarkers for PD in biological fluids...” I wonder why the authors did not mention and discuss the studies on saliva which are recently raising lots of interest.
Answer: The search (Supplementary file 1) performed to build this review included the term “saliva”, but none of the retrieved results presented data on autophagic- and lysosomal-related biomarkers on this fluid. Additionally, the authors performed an additional search and were not able to find any articles on the topic. In order to make the readers aware of the lack of data regarding saliva biomarkers related to the autophagic-lysosomal pathway, the authors added a couple of sentences in the discussion (line 430): “Additionally, current studies on ALP-associated biomarkers for PD lack two non-invasive biosamples used for the diagnosis and monitoring of neurological diseases, namely saliva [130] and urine [131]. It would be interesting to study if those biospecimens can be used to detect some of the biomarkers discussed here.”
2) Line 68: “… or viral infections [14].” The cited article does not discuss any viral infections then authors should amend the sentence, maybe saying something related to the prion-proteins that behave as infection agents but aren’t (that is mentioned in the cited paper).
Answer: The authors have amended the sentence by adding proper citations to all information. More specifically, reference on a review dealing with the role of endoplasmic reticulum and autophagy (Reference 15; PubMed ID: 26389781) and a reference on autophagy and infection, which includes viral infections (Reference 16; PubMed ID: 24064518) have been added.
3) Line 108: “… GBA-PD patients usually manifest an earlier onset and a more severe course of the disease…” which disease the authors are talking about?
Answer: The authors acknowledge that this sentence could lead to a confusion. We modified it to: “Third, GBA-PD patients usually manifest an earlier onset and a more severe course of the parkinsonism compared to those PD patients not carrying GBA mutations, in particular with greater cognitive decline, faster progression and increased risk of mortality”.
4) They stated in line 104 “The relevance of PD associated with GBA mutations (GBA-PD) resides in multiple factors” but not mention the evidence of association nor cited any reference therefore it is not clear what are they referring to.
Answer: The sentence “The relevance of PD associated with GBA mutations (GBA-PD) resides in multiple factors” is a preface to the following four sentences, starting with “First”, “Second”, “Third”, and “Fourth”, which describe (and give references) the factors that highlight the relevance of GBA mutations in the context of Parkinson’s disease.
5) Why authors discussed the data on sSIN related only to GBA-PD and did not at least mention the other groups? And only mention the total aSIN despite the oligomeric and phosphorylated forms?
Answer: The authors are aware that the text focuses only in alpha-synuclein as a biomarker for GBA-PD cohorts. As stated in the first paragraph of the “a-syn levels” chapter, the decision was taken to avoid overlap with other reviews that devoted all their efforts on reviewing specifically the topic of a-syn for PD biomarker development, including GBA-PD and other groups of PD. The authors refer to specific references for additional detailed information on this topic (Refs 11 and 21). Instead, the authors decided to highlight exclusively studies focused on GBA-PD since the chances of finding consistent data to proof the usefulness of alpha-synuclein as a biomarker could be higher than with other groups. This assumption was made based on two evidences: (i) protein quality control mechanisms such as autophagic clearance are essential to regulate the levels and accumulation of alpha-synuclein, and (ii) the autophagic-lysosomal function is compromised in GBA-PD cases. Regarding the second part of the question, the available data presented in this review on alpha-synuclein in GBA-PD, which fulfills the biomarker criteria, is focused on total levels, together with oligomeric and dimeric forms, and are the ones mentioned in the text (paragraph starting in line 325).
6) Line 418-420: “…because of the importance of the ALP pathway for PD independently of GBA mutations, one could hypothesize that finding the biomarkers that allow to predict the conversion of GBA carriers to PD may also allow us to reveal those who will develop sPD…” again I found this statement more a speculation than a conclusion with scientific basis.
Answer: The authors, by writing “one could hypothesize”, referred to the following statement as a speculation rather than a scientifically-based conclusion.
7) Moreover, talking about the alterations in lipid metabolism and endo-lysosomal pathways that is actually the topic of this review it should be interesting – and I wonder why authors did not mention that – to discuss the lipid droplets involvement if there is.
Answer: The authors agree that, considering the relevance of lipid droplets in autophagy, it is worth mentioning the lack of articles studying lipid droplets as autophagic- and lysosomal-related biomarkers. Hence, a sentence has been added in the discussion (line 449): “Moreover, some ALP-related parameters, such as lipid droplets [132], have not been explored yet.”
8) At a certain point in the narrative start appearing several acronyms which, in particular for a reader that it is not properly involved in this scientific area, are pretty hard to follow and understand especially cause the lack of any information useful to. For example: LRRK2, PARKN, PRKN and several mutations that are mentioned but without any explanation where they are and provoke therefore I found confusing and disturbing in the reading, it may be better to avoid them unless simple useful info are reported.
Answer: The authors have added a table (Table 1) with the commonly found mutations associated with PD, together with their involvement in the autophagy-lysosomal pathway. In this table, the symbol and the name of the currently known familial genes involved in Parkinson’s disease is described. The authors hope that with this table, the presence of the above mentioned acronyms in the text will not be confusing and disturbing anymore.
9) Agreeing that the GBA-PD can be considered and represents a good model which can to tell us something more related to the ALP involvement in the PD, but the title and the abstract in this context sound a little bit tricky and pretentious especially when it is stated that “…evidence exists that the impairment of the autophagy-lysosomal pathway underlying pathophysiology…” since the pathophysiology is due to the GBA mutations then the conclusion seems to be sort of biased.
Answer: When the authors refer to the sentence “…evidence exists that the impairment of the autophagy-lysosomal pathway underlying pathophysiology…” in the abstract and in the conclusion section, they are not referring exclusively to the evidences of altered autophagic-lysosomal pathway in GBA-PD patients. As indicated in the introduction, ALP dysfunction is common to several neurodegenerative diseases (line 74) and the impairment of ALP in PD is supported by several evidences that are not restricted to GBA-PD: (i) ALP represents one of the major routes for the intracellular degradation of a-syn, which accumulation is common to sPD and genetic PD cases (line 77); (ii) genetic studies reported familial PD genes involved in ALP and common genetic variants in lysosomal genes associated with the risk of PD (line 96); (iii) altered GCase activity has also been reported in other genetic cohorts (i.e. LRRK2, PRKN and SNCA) and even in sPD patients (line 125). Thus, the authors believe that there is scientific evidence to support the fact the ALP impairment underlies PD pathophysiology, in general. Another idea is that the authors highlight the GBA-PD group as a relatively frequent, well defined and stratified cohort that is currently in the spotlight as one of the most promising prodromal PD cohorts to use in future clinical trials for disease-modifying therapeutic interventions (lines 125 and 146). The in-depth study of this cohort, as the reviewer indicates, represents a good model which can tell us something more related to the ALP involvement in PD, and importantly, as discussed in line 128 “… biomarkers monitoring the function of the ALP and therapeutic approaches modulating this pathway could potentially be also of relevance to a wide range of PD patients and subjects at-risk”. However, because this is mere speculation, the authors refer to it in the conditional verb tense and is subject to the results from future studies.